# Identifying Learners’ Interaction Patterns in an Online Learning Community

**DOI:** 10.3390/ijerph19042245

**Published:** 2022-02-16

**Authors:** Xuemei Wu, Zhenzhen He, Mingxi Li, Zhongmei Han, Changqin Huang

**Affiliations:** 1School of Information Technology in Education, South China Normal University, Guangzhou 510631, China; wuxuemei@m.scnu.edu.cn (X.W.); zzhe@m.scnu.edu.cn (Z.H.); 2School of Foreign Studies, South China Normal University, Guangzhou 510631, China; mingxili72@163.com; 3Key Laboratory of Intelligent Education Technology and Application of Zhejiang Province, Zhejiang Normal University, Jinhua 321004, China; zm.han@zjnu.edu.cn

**Keywords:** collaborative reflection, emergent roles, interaction pattern, deep learning, online learning community

## Abstract

The interactions among all members of an online learning community significantly impact collaborative reflection (co-reflection). Although the relationship between learners’ roles and co-reflection levels has been explored by previous researchers, it remains unclear when and with whom learners at different co-reflection levels tend to interact. This study adopted multiple methods to examine the interaction patterns of diverse roles among learners with different co-reflection levels based on 11,912 posts. First, the deep learning technique was applied to assess learners’ co-reflection levels. Then, a social network analysis (SNA) was conducted to identify the emergent roles of learners. Furthermore, a lag sequence analysis (LSA) was employed to reveal the interaction patterns of the emergent roles among learners with different co-reflection levels. The results showed that most learners in an online learning community reached an upper-middle co-reflection level while playing an inactive role in the co-reflection process. Moreover, higher-level learners were superior in dialog with various roles and were more involved in self-rethinking during the co-reflection process. In particular, they habitually began communication with peers and then with the teacher. Based on these findings, some implications for facilitating online co-reflection from the perspective of roles is also discussed.

## 1. Introduction

Collaborative reflection is a process of embracing the critical thinking of individuals and groups to achieve a deep understanding of knowledge [1]. Thinking together can allow learners to open themselves by sharing experiences, accepting feedback, and providing justifications, which can alleviate their loneliness, promote their collective knowledge construction, and reinforce the healthy development of the learning community, especially in an online environment [2,3]. A high level of co-reflection represents the quality of individual reflection in collaborative engagement, indicating that the participant can identify gaps in collective knowledge, maintain the continuous improvement of conceptual depth, and form deep and complex views about theories or concepts [4]. Online discussion data, especially learner posts, contain rich implicit information that help to understand learners’ co-reflection thinking [5]. Accordingly, many researchers have attempted to identify the co-reflection level of learners based on interactive texts using various methods such as content analysis methods, machine learning-based methods, and rule-based methods [6].

However, most scholars have stated that it is difficult for learners to reach a high-level of co-reflection in an online learning community. Learners are used to describing situations or sharing their own experiences, rather than thinking deeply through discussions with others, thereby obtaining an undesirable reflection level [7]. Therefore, some assistance should be provided to support learners’ online co-reflection. In general, existing research has contributed to this through three aspects. One is to design scaffolding to support collaborative reflection [8]. The second is to investigate the possible reasons why learners have undesirable responses in relation to co-reflection [9]. The third is to provide some implications by exploring the co-reflection patterns of learners in terms of cognition, emotion, behavior, etc. [5,10,11].

Several researchers have discussed the behavioral patterns implied in students’ collaborative discourse [5,11]. For example, Liu et al. analyzed the level and the evolution of teachers’ reflective thinking in online collaborative learning activities [5]. Although these studies characterize the learners’ collaborative reflective behavior, they ignore the fact that interactions are dynamic, and that learners usually interact with different members in an online learning community [12]. More importantly, a learner’s reflective behavior and level can be affected by working with different community members, especially by those who are more authoritative, influential, and capable [9]. However, it is unclear with whom learners prefer to interact in an online learning community and how this affects their co-reflection levels.

Roles have been regarded as a key characteristic that distinguishes an individual’s identity from the others in a group. Notably, individuals’ emergent roles are established spontaneously in line with their unique interaction behavior in the online community [13]. Furthermore, the participation behavior of an individual and the group can be influenced if learners work with different members playing various roles [14]. In particular, the roles teachers play in discussions can impact the participation and perception of students [15]. As stated by Ouyang and Scharber, the teacher plays different roles such as guide, facilitator, observer, and collaborator at different times in the online discussion, which influences the learners’ interactions and participation in an online learning community [16]. Moreover, a teacher’s facilitation behavior, especially when assuming a more collaborative rather than instructive facilitation manner, plays an important role in promoting the learners’ reflective thinking level and social dialog during collective reflection [17]. Additionally, the time and order of the roles’ emergence might play an important part in the learners’ engagement and contributions [18,19]. Particularly, individuals’ strong identification with others in the learning community is a critical factor in changing their reflective behavior and thoughts [20]. For instance, the attractiveness, fairness, and similarity perceived by the followers of an influencer in social interactions will lead them to imitate the influencer’s behavior [21]. Pozzi pointed out that despite the diverse roles played by learners, most of them develop a higher level of individual and group knowledge building [22]. Furthermore, it has been found that an individual’s deep-level knowledge inquiry could trigger peer interaction and further advance group knowledge construction [23]. In light of this, it is necessary to clarify the sequential pattern of when and in which roles learners with different co-reflection levels habitually dialog within an online learning community.

This study aims to contribute to the existing research by revealing the interaction patterns of learners with different co-reflection levels, especially considering the roles learners and teachers play in an online learning community. The findings will be helpful for instructors within the online learning community in assisting learners to improve their co-reflection level from the perspective of roles.

## 2. Literature Review

### 2.1. Collaborative Reflection

Collaborative reflection refers to a collaborative critical thinking process involving cognitive and affective interactions between two or more individuals who explore their experiences to reach new intersubjective understandings and appreciations [1]. In this way of thinking, we can learn that individual and group cognition are intertwined in the co-reflection process, and learners’ experience and knowledge can be internalized through dialog with others [24]. Adhering to social constructivism, which was proposed by Vygotsky [25], learners have the opportunity to engage in meaning-making through interaction with their peers and teachers. More importantly, some scholars have claimed that healthy communication with peers such as through equal stance, positive attitude, and an open mind could facilitate learners’ cognitive engagement [26,27]. Based on these findings, we can see that effective communication with others in an online learning community plays an important role in deepening learners’ collective reflection and enables them to understand knowledge profoundly.

However, unfortunately, learners cannot always benefit from these dialogs because the signals they receive are discrepant in an online learning community [2]. For example, a cohort of learners stated that they were inspired by their instructor [9]. In contrast, some students felt that they were able to think more closely about the topic with the instructor’s support while reflecting more deeply about it by self-thinking or discussing it with their fellows [28]. Notably, more detailed explanations have been contributed by other researchers. According to the findings of Hou [29] and Watanabe [30], learners are willing to share more experiences and opinions with those members who are in the same situation as themselves and share numerous ideas regardless of their proficiency. Moreover, several scholars have emphasized that students can benefit greatly by providing more feedback to others than by receiving evaluations [31].

Consequently, some valuable clues can be detected from these contributions. On the one hand, collaborating with different agents in a critical thinking process might be a key factor that causes inconsistent co-reflection levels among learners. On the other hand, several key characteristics of learners, such as authority, influence, and popularity, may play an important role in progressing their co-reflection. However, an ambitious elaboration is hard to find in the existing research. Taking these into consideration, this study investigates the question of which co-reflection levels of learners often interact with each other in an online learning community.

Additionally, to obtain valuable information on co-reflection from the large-scale online interactive text data, many researchers have focused on developing advanced approaches. Particularly, machine learning methods such as the support vector machine method [32], random forests [33], and the Bayes method [5] have been applied to analyze learners’ reflection levels automatically. However, these methods have some limitations, such as being labor-intensive and time-consuming. More importantly, deep learning possesses a multi-layer network structure and a powerful learning capability, and it is considered to be the most powerful technique for text mining [34]. In education, Zou, Hu, Pan, Li, and Liu [35] proposed a text classifier based on bidirectional encoder representations from transformers (BERT) to analyze forum posts in terms of social presence. Han, Huang, Yu, and Tsai [36] employed long short-term memory (LSTM) to identify online learners’ various epistemic emotions. Huang et al. [20] combined BERT and LSTM to analyze learners’ reflective texts in the online learning community. Although the significance of these advanced approaches in education settings has been well-documented, fewer efforts have been made to analyze online interactive reflection texts in order to reveal learners’ online co-reflection states. Therefore, the present study attempts to apply deep learning techniques to analyze online co-reflection texts and further identify the co-reflection level of learners in an online learning community.

### 2.2. Emergent Roles in an Online Learning Community

Adhering to the theory of social identity [37], individuals can establish their identities in a group through the affective perception of the community. Their behaviors and thoughts can be modified if they identify with other members in the group [38]. More importantly, roles represent the individual’s unique identification or behavior pattern in the group and play an important part in collaborative learning. Generally, there are two categories of roles: scripted roles and emergent roles. Scripted roles are prescribed to standardize learners’ participation behavior and to facilitate their knowledge construction, to which previous scholars have given much attention [14]. Nevertheless, assigning roles, to some degree, may prevent students from solving problems naturally [39]. As expected, emergent roles, which develop spontaneously in response to a collaborative activity, provide meaningful information about an individual’s behavior pattern embodied in the group discourse [14]. For example, Ouyang and Chang identified six emergent roles based on students’ social participatory behavior. The peripheral refers to students who usually perform inactively in social engagement and make the least commitment to knowledge building; they usually receive few responses and little feedback from their peers [23]. Particularly, it is worth noting that learners are inclined towards more consistent behavior over time once they have established their role [40]. Thus, external support should also be provided to maximize learning gains.

Several scholars have highlighted that the role teachers play, such as being the facilitator, guide, etc., have positive effects on students’ reflective thinking and engagement in the online learning community [41,42]. In particular, the teacher as a guide clearly defines reflection tasks and requirements, monitors the learning process, and offers suggestions, which can increase the learners’ emphasis on reflection and engage them in critical thinking [43,44]. Furthermore, the teachers’ facilitation has a significant effect on learners’ behavioral and cognitive engagement in online discussions [42]. The facilitator provides feedback and scaffolding for learners’ reflective writing, which is helpful in improving their reflection quality [44]. Moreover, receiving evaluations and thinking deeply about others’ responses to their own comments may have a positive influence on students’ critical thinking [45]. However, few researchers have explored the interactions between teachers and students while considering their various roles and their impact on learners’ co-reflection.

Numerous researchers have attempted to identify different emergent roles based on various methods, such as content analysis, SNA, and combinations thereof. For example, Eubanks, Palanski, Olabisi, Joinson, and Dove [46] assessed team roles (e.g., project coordinator, implementer, completer-finisher, etc.) using content analysis based on the behavior checklist. Marcos-García, Martínez-Monés, and Dimitriadis [41] represented the fine-grained emergent roles of teachers (e.g., guide, facilitator, observer) and learners (e.g., leader, animator, peripheral, etc.) in online collaboration based on a set of properties and SNA indexes. Swiecki and Shaffer [47] combined epistemic and social network analyses to identify the particular roles of team members, such as those of a commanding officer, tactical action officer, and warfare supervisor. A novel and automatic approach, group communication analysis (GCA), was proposed by Dowell, Nixon, and Graesser [48] to detect emergent social roles such as drivers, influential actors, and lurkers. Furthermore, Dowell and Poquet [49] analyzed emergent roles (e.g., followers, influential actors, hyper posters, etc.) in digital environments based on the GCA and SNA approaches. Overall, social network analysis, as a fairly mature method, has mostly been used to extract the network structure and identify learners’ roles within complex social relations. However, several researchers recommend that more comprehensive information (e.g., reference relations) should be considered to accurately reveal the communicative relations between learners [50]. Hence, this study uses SNA to identify the learners’ roles that have emerged in the co-reflection process according to their reference relations and technological relations.

### 2.3. Online Learners’ Interaction Patterns

Exploring the interaction pattern of online learners is helpful for disclosing their subtle learning characteristics that involve cognition, emotion, and behavior [51]. Among these studies on learning behavior, several researchers explored the interaction structure of learners’ collaborative knowledge construction. For example, Zhang, Skryabin, and Song [52] examined the patterns and evolution of relationships formed in the context of an MOOC discussion forum. Tao and Zhang [53] revealed students’ dynamic knowledge-building structure based on SNA, content analysis, and narrative approaches. In addition, Wise and Chiu [54] illustrated temporal patterns of knowledge construction in a role-based (e.g., starter, wrapper, questioner, synthesizer, etc.) online discussion. Particularly, some scholars focused on the temporal characteristic of students’ roles in the collaborative learning process. A study conducted by Farrow, Moore, and Gašević [19] explored the impact of the order in which students took on each role, namely those of the research expert and practicing researchers, on the quality of their contributions to other threads. Similarly, De Wever, Van Keer, Schellens, and Valcke [18] examined the introduction of the starter, summarizer, moderator, theoretician, and source searcher roles, and demonstrated that these roles should be introduced at the start of the discussions and fade out towards the end. Ouyang and Chang [23] explored the relationship between students’ social participatory roles (e.g., leader, starter, influencer, mediator, regular, and peripheral) and cognitive engagement levels in an online discussion and underlined that the change of individual roles can affect the behavior of knowledge inquiry, trigger peer interaction, and influence the construction of group knowledge.

In summary, these studies imply that the interactive network of the online learning community is dynamic, and that the timing of the interaction with various roles, such as guide, facilitator, influencer, starter, etc., may affect students’ participation behavior and learning performance differently. However, the temporal interaction pattern regarding which roles students use to dialog is still unclear.

Due to its power to examine the probability of one behavior occurring after another and its statistical significance, lag sequential analysis (LSA) has been prominent in the literature for uncovering the temporal interaction pattern. For example, Huang, Han, Li, Jong, and Tsai [55] investigated students’ interaction patterns regarding dynamic learning sentiments using content analysis and LSA methods. LSA methods were applied by Tlili et al. [56] to understand the behavioral patterns of online learners from three cultures. Given the purpose of this study, LSA was employed to analyze the interaction patterns of learners’ emergent roles in the online co-reflection process.

### 2.4. Research Questions

Many studies have contributed insights on online co-reflection, but most depend on the manual coding method. Few studies have been conducted to identify the learner’s online co-reflection level based on machine learning techniques. Furthermore, although some information about the relationship between learners’ roles and co-reflection levels can be obtained from the existing literature, little is known about the emergence of learners’ roles in a co-reflection process, and it remains to be seen how learners sequentially interact with each other depending on their roles. To fill these gaps, the current study aims to explore the differences in the sequence patterns of emergent roles among learners with different co-reflection levels. More specifically, by integrating deep learning, SNA, and LSA methods, this study attempts to answer the following three questions:(1)What is the status of learners’ co-reflection in an online learning community (assessed by deep learning)?(2)Which roles of learners emerge in the process of co-reflection in an online learning community (analyzed by SNA)?(3)What interaction patterns of emergent roles do learners with different co-reflection levels demonstrate (processed by LSA)?

## 3. Methodology

### 3.1. Research Context and Participants

A total of 42 postgraduates (32 females and 10 males, with an average age of 24.4) were required to take the course on fundamental educational technology theory to strengthen their knowledge on the application of technology in education. The proportions and profiles of the participants were consistent with the actual distribution across the university. All students were enrolled in the same major in educational technology, and they had equivalent professional knowledge as well as rich experience in online learning and technology application, so all of them were invited to participate in this research on a voluntary basis.

Supporting previous findings that these roles—namely guide, facilitator, and observer—play an important part in learners’ interactions and reflection from different aspects [42], this study assigned them to three teachers in this course, respectively. All of the teachers were trained to play the three roles before the class. As a guide (G), the teacher’s main task was to design collaborative reflection topics, devise the task requirements, monitor the entire co-reflection process, and give a demonstration [16]. The facilitator (F) was responsible for providing support to meet the needs of the learners, including answering questions and providing resources, tools, and materials [41]. Thus, it was not necessary for the facilitator to exert a strong influence, but rather a medium level of participation and mediation was appropriate because the intensity of these actions depended on the activity of the students [42]. The teacher assigned to an observer (O) role was required to ask students to provide reliable evidence and explanations for their views about learning topics [57]. Thus, the observer would not engage in high participation, influence, and mediation. Furthermore, it should be noted that none of the teachers interfered with students’ interactions intentionally. That is to say, learners had the right to decide when and with whom they were willing to interact, and they would constitute their roles in line with the interaction pattern embodied in the collaboration with other members.

To support collaborative reflection activities, it was recommended that all learners engage in discussions with other members based on the discussion forums set up in a Moodle-based online learning platform (Liru Cloud Classroom). The platform was developed to support the teaching of all academic staff and the learning of students in a research university in south China. Thus, they were all familiar with the platform. Furthermore, this platform provided a favorable environment to support learners’ co-reflection. On the one hand, teachers could release course-related learning materials, discussion topics, and tasks and monitor the progress of the learners’ learning and discussion. Therefore, the platform could better support teachers in guiding, assisting, and monitoring the learners’ learning process. On the other hand, learners could choose to share their experiences, post reflection logs, and communicate with others using computers or mobile devices. Before starting the course, they were all trained on the operation of the platform to ensure that they would not be hindered due to technical issues in the co-reflection process. During the course, each student participated in seven co-reflection topics within the 12 weeks in exchange for credits. The specific themes and processes of online co-reflection are shown in Figure 1. Finally, 39 learners were considered in this study after excluding those students who did not participate in any interaction. A total of 1531 posts were gathered.

### 3.2. Coding Scheme

#### 3.2.1. The Coding Scheme for Collaborative Reflection Levels

To examine the learners’ co-reflection status in an online learning community, we adopted the coding scheme for co-reflection levels developed by Lei and Chan [4]. As shown in Table 1, there are nine strategies for assessing learners’ co-reflection, which can be grouped into three levels: low, medium, and high. At the low level, students simply share information by listing and copying others’ notes, making short summaries, or interpreting others’ notes in different ways. At the middle level, learners’ co-reflection is concentrated on knowledge construction, including those strategies of “question-based discussion”, “constructive use of information”, and “intertwined question explanation”. At the high level, learners might participate in a meta-discourse process involving “meta-cognition”, “meta-theory”, and “meta-conversation”.

Two independent coders were invited to participate and were trained in the coding scheme to code the reflective posts. To train an optimal model for co-reflective level identification, a large-scale and labeled co-reflective text is required. Since the data gathered in the target course were insufficient to train this model, as suggested by Huang et al. [58], an additional 10,381 pieces of co-reflective text data were collected from other courses. Notably, the students in these courses were all enrolled in the same major in educational technology, and all of them were enrolled in courses related to the major. These students participated in similar co-reflection activities in these courses based on the “Liru Cloud Classroom” platform, and each student engaged in co-reflection topics posted by the teacher during a semester. Therefore, these co-reflective texts were included and prepared as the training dataset to obtain an optimal classification model for the co-reflection level. A complete dialog with the same person on each topic was regarded as an analysis unit [20]. Each unit was coded into one of the three co-reflection levels by the two coders, while those that did not belong to any of the nine categories were coded as “other”. A consensus was reached through discussions to resolve any discrepancy in the coding results. The Kappa value was 0.746 (*p* < 0.01), suggesting a high level of agreement between the coders.

#### 3.2.2. The Metrics of Emergent Role Identification

To identify the emergent roles of learners in a co-reflection process, a measurement proposed by Ouyang and Chang [23] was utilized. As described in Table 2, six roles—namely leader, starter, influencer, mediator, regular, and peripheral—were the target categories in the present study. All were classified based on a series of properties (participation, influence, and mediation) and SNA indexes (e.g., indegree, incloseness, betweenness, etc.). That is to say, each role represents the learner’s interaction pattern with regard to participation, influence, and mediation [41]. Participation refers to the intensity of the learners’ engagement in activities. Influence is related to the influence and popularity of a student. In addition, Mediation indicates the individual’s control over the flow of information or resources in a social network. As stated before (in Section 2.1), students who collaborate with others with these characteristics may impact their reflective performance. In terms of SNA indexes, the metric of degree includes outdegree and indegree. Outdegree is a measure of how many outgoing interactions are made by an individual; that is, how many times the individual actively communicates with others [41]. Indegree is a measure of how many incoming links an individual receives, reflecting the level of the individual’s popularity [41]. The metric of closeness, which includes incloseness and outcloseness, measures how close a participant is to others on average, representing the efficiency of the participant in exchanging information with others [23] via incoming or outgoing links [41]. Moreover, betweenness measures the mediatory effect an actor exerts over others in a network [23].

More specifically, a leader student has at least two characteristics with a high level of participation, influence, and mediation; these students can explain others’ ideas in detail and receive their feedback. In contrast, students who have at least two characteristics with a low level of participation, influence, and mediation (at least two of them) are considered peripheral students. A regular student offers middle-level contributions in participation, influence, and mediation. Furthermore, starters are prominent in high-level participation since they are prone to actively initiating social interactions. Mediator students distinguish themselves from others by their excellent mediation effect; these students contribute to bridging two members in the co-reflection process. The influencer identifies those students with a high-level of influence but with medium or low levels of participation and mediation.

### 3.3. Data Analysis

Three methods were adopted in this study (see Figure 2). First, the deep learning approach that integrates a lite BERT (ALBERT) and LSTM was performed to classify the co-reflection level of each learner. Second, the SNA method was used to identify learners’ emergent roles in online co-reflection. Third, LSA was conducted to examine the interaction patterns of emergent roles among learners with different co-reflection levels.

#### 3.3.1. Classifications of Collaborative Reflection Levels using Deep Learning

Given that learners’ reflection is implicit and the reflective text is temporal and large-scale, the present study combined ALBERT and LSTM techniques to automatically identify learners’ co-reflection levels. More specifically, ALBERT, which shows excellent performance on the understanding of semantic information even when trained with the limited data [59], was used to extract features of the online co-reflection text in this research. In addition, LSTM, which is suitable for analyzing sequential information, was employed to capture the sequential feature of learners’ interactive reflection text [60].

Four phases were needed to determine the learners’ co-reflection levels based on the proposed approach. In phase 1, we defined the co-reflection level codes. A total of 11,141 reflective posts were coded in chronological order to produce a training dataset. In phase 2, we built the classification models. The Word2vec, BERT, random forests (RF), and TF-IDF (term frequency–inverse document frequency) methods, which have been commonly used in text classification tasks, were combined. In phase 3, we trained and selected the optimal model. The ten-fold cross-validation was implemented to determine the optimal classifier. More specifically, all training data were equally divided into ten parts over ten iterations, nine parts of which were used to train the model, and the remaining were the testing data for validation [61]. Moreover, four metrics, namely accuracy, precision, recall, and the harmonic mean of precision and recall (F1), were used to evaluate the performance of these algorithms. As shown in Table 3, the proposed method demonstrates a significant improvement over the previous models. In phase 4, we applied the trained classifier to assess 39 learners’ co-reflection levels.

#### 3.3.2. Emergent Role Identifications Using Social Network Analysis

To identify the emergent role of learners, the communicative relations between each learner should initially be determined. Two types of relations suggested by Engel et al. [50] were considered in this study. One is the connections established by learners through commenting, replying, etc. The other relation is defined by the personal references to other participants that have made contributions. After depicting these relationships, the SNA was implemented to examine the centrality indexes using the UCINET 6.186 software. Accordingly, the levels of learners’ participation, influence, and mediation were determined, and the emerging role of each learner was finally identified.

#### 3.3.3. Interaction Patterns of Learners’ Emergent Roles Using Lag Sequential Analysis

LSA was carried out to reveal the sequence patterns of learners with different co-reflection levels. Particularly, the importance of learners’ self-internalization [24], “self”, as an independent concept, was considered in this research. If a student dialogs with himself, it not only means that he can actively construct knowledge but also implies that other members can influence his self-thinking in the process of co-reflection.

As a result, ten roles of teachers and learners were analyzed to reveal the interaction pattern using three steps. First, the roles that each learner chose to interact with were coded in chronological order. Second, the adjusted residuals (Z-score) of the connection between each sequence was calculated using GSEQ 5.1. Notably, if the Z-score was equal to or higher than 1.96, this indicated that a given sequence had reached a significant level [62]. Third, the sequential transfer diagram was drawn to visualize the changing pattern between diverse roles.

## 4. Results

### 4.1. Status of Different Co-Reflection Levels among Learners

The best performing text classifier (ALBERT and LSTM) was used to evaluate each learner’s co-reflection level. As a result, out of the 39 learners, 8 (20.51%) had a low co-reflection level, 19 (48.72%) performed at a middle level, and 12 learners (30.77%) achieved a high level. This indicates that most learners engage in deeper collaborative reflection in the online learning community.

To examine the interactive discourse among learners with different co-reflection levels, we used a set of word clouds (see Figure 3). Generally, all learners paid more attention to the core ideas of all topics, such as technology, learning, education, and teaching. However, some subtle differences in their collaborative discourse and strategies were noted. Learners at the high co-reflection level (see Figure 3c) focused on sharing collective knowledge and formulating insightful views on theories or concepts. Learners at the middle level completed tasks, solved problems, and engaged in knowledge acquisition (see Figure 3b), whereas learners at the low co-reflection level responded more to conceptual understanding based on existing references or experience, but lacked an in-depth construct of these concepts (see Figure 3a).

### 4.2. Emergent Roles in the Online Co-Reflection Process

A total of six roles were identified among the 39 learners based on the SNA. As depicted in Table 4, 25 learners took on regular (*n* = 13, 33.33%) and peripheral (*n* = 12, 30.77%) roles. Six learners (15.33%) played the leader role, three took the role of influencer (*n* = 3, 7.69%), and three the role of mediator (*n* = 3, 7.69%). Only two starters (5.13%) participated in co-reflection. In other words, most learners in an online community took an inactive stance on collaborative reflection.

Furthermore, according to the results of ANOVA and the Kruskal–Wallis test, learners in six roles showed statistical significance on outdegree, outcloseness, indegree, incloseness, and betweenness (see Table 5). In particular, based on the post hoc tests, influencers achieved a high score for indegree (mean = 56.33, S.D = 4.03) and inclossness (mean = 73.26, S.D = 1.88), but poor in outdegree (mean = 34.00, S.D = 5.72), outcloseness (mean = 58.63, S.D = 1.78), and betweenness (mean = 25.56, S.D = 7.93). This demonstrated that these learners had high social impact, whereas they had low participation and mediation. With regard to starters, they achieved a high score for outdegree (mean = 65.00, S.D = 1.00) and outcloseness (mean = 72.62, S.D = 1.93) despite having a middle-level betweenness (mean = 36.73, S.D = 0.40) and a lower score for indegree (mean = 13.00, S.D = 1.00) and incloseness (mean = 57.37, S.D = 1.20). This indicates that these learners participated actively in the co-reflection. Moreover, mediator learners scored high for betweenness (mean = 54.49, S.D = 6.63), whereas they had a middle score for other indicators. This suggests that they had the power to bridge the interaction among different community members. Furthermore, leaders showed high involvement for degree, closeness, and betweenness, whereas regulars demonstrated a middle commitment in all these measures. Peripherals had the lowest scores.

### 4.3. Interaction Patterns of Learners’ Emergent Roles according to Their Co-Reflection Levels

The interaction patterns among high, middle, and low co-reflection level learners were examined by using a set of sequential analyses. For learners with a low co-reflection level, as reported in Table 6, the rows show the starting roles. The columns represent roles that occur immediately after the end of the start. In light of the Z-score, Figure 4 visualizes a transition diagram. As shown, there are five significant sequences, namely “guide → self”, “facilitator → self”, “self → starter”, “observer → mediator”, and “mediator → starter”. Generally, learners at the low co-reflection level tended to interact with the teacher first and then with other peers, but only the guide and facilitator were able to move them to self-reflection.

Similarly, the adjusted residuals table of sequential analyses (see Table 7) and the visible transition diagram of middle-level learners were obtained (see Figure 5). With this result, two interaction paths were adopted by these learners: “influencer → influencer → observer → self → guide → starter”, and “influencer → influencer → observer → self → facilitator → peripheral”. Of these, the path of “influencer → influencer → observer → self” showed that these learners were inclined to communicate with the influencer many times initially, and then with the observer. This implies that the influencer’s point of view may inspire middle-level learners to a greater extent, whereas the observer plays an important role in enhancing their self-thinking. Interestingly, two paths after self-reflection were observed. One is that learners interact with the guide, followed by the starter (“self → guide → starter”), and the other is that learners make contact with the facilitator, then with the peripheral (“self → facilitator → peripheral”). This suggests that middle-level learners will communicate with others after solving their own questions or responding to the teacher’s request. Consequently, learners with a middle co-reflection level placed more emphasis on the voice of influencers rather than other peers, and they rarely think again in this process unless the observer asks them to.

Comparatively, according to the adjusted residual table (see Table 8) and interaction transition diagram (see Figure 6), ten significant sequences were found. Of these, the sequences of “facilitator → self”, “guide → self”, “observer → self”, “starter → self” indicate that both teachers (guide, facilitator, and observer) and starters can directly promote students’ further self-thinking. Remarkably, the bi-directional sequences, “self ↔ guide”, “influencer ↔ observer”, and “self ↔ starter” clarified that high co-reflection level learners made frequent interactions between teachers and peers, and they subsequently internalized these ideas learned from others. Furthermore, the significant interaction paths of “peripheral → mediator → facilitator → self” and “influencer → observer → self” showed that high-level learners tend to engage in dialog with more peers before they engage in self-thinking, which is different from learners at a low or middle level.

In analyzing the interaction patterns according to the learners’ three co-reflection levels, some similarities and differences can be further clarified. Regardless of whether learners are at the high, middle, or low co-reflection level, interacting with teachers can directly motivate their self-reflection. In particular, learners at the high co-reflection level can be inspired by the observer, guide, and facilitator, while learners at the low-level can be influenced by the guide and facilitator. In contrast, learners at the middle-level only engage in self-reflection after interacting with the observer. Moreover, there are two differences in the interaction pattern between low-level learners and upper-middle level learners. On the one hand, low-level learners liked to interact with teachers at the outset, whereas middle and high co-reflection level learners preferred to engage in dialog with teachers after peers (e.g., influencer). On the other hand, learners with low co-reflection levels interacted with peers (e.g., starters, mediators) after self-reflection, which was different from middle and high co-reflection level learners who further interacted with teachers.

Furthermore, three differences between middle-level and high-level learners should be highlighted. First, although both had a tendency towards communion with influencers initially and then with the observer, the difference is evident; that is, high-level learners tended to interact back and forth between the influencer and the observer, whereas middle co-reflection level learners tended to engage in dialog with the influencer multiple times, and then with the observer. Second and another difference worth noting is that the sequence of “self → guide → starter” was irreversible for the middle-level learners. However, the sequence for high co-reflection level learners was circular (“self → guide → starter → self”). That is to say, high-level learners would reflect again after interacting with the guide and starter, which allows them to obtain a higher level of co-reflection. Third, high co-reflection level learners communicated with “peripheral → mediator → facilitator → self” before self-reflection, whereas the middle-level learners communicated with the facilitator and peripheral learners after self-reflection (“self → facilitator → peripheral”). This indicates that middle co-reflection level learners may be persuaded by those learners with more influential opinions, whereas high-level learners might not.

Additionally, the results of the chi-square analysis (as illustrated in Table 9) further revealed the differences in the interaction frequency of different roles among high, middle, and low co-reflection level learners (χ^2^ = 47.65, *p* < 0.001). In addition, a posteriori comparison was performed to examine the differences among these learners in detail. In terms of the guide, influencer, peripheral, and self, significant differences were found between high and low co-reflection level learners, whereas a significant difference was found between the high and middle-level learners for the role of starter. Therefore, it can be concluded that interacting with learners in diverse roles (e.g., guide, influencer, starter, etc.) and engaging in more self-reflection in this process may enable learners to be more deeply involved. Combined with the findings of the interaction pattern above, we can see that it is largely the difference in interaction sequence that results in learners having diverse levels of co-reflection.

## 5. Discussion

This study investigated the interaction patterns of learners with different co-reflection levels considering their diverse roles in an online learning community. More specifically, this study answered three questions detailed in Section 2.4. In this subsection, we discuss how our findings answer these questions.

Regarding the first research question (RQ1), which is related to the status of learners’ co-reflection in an online community, the findings reveal that learners can generally achieve a middle-to-high level of co-reflection, and that they engage in topic-related thinking. This is not consistent with the previous findings that most students only discuss what happened or talk about something unrelated [9]. Some possible reasons can be explained from the perspective of the community of inquiry. More specifically, given that teaching plays a vital role in fostering learners’ cognitive and social presence [63], we believe that the role of teachers has an impact on learners’ engagement in co-reflection. Moreover, we agree with Yılmaz ’s [64] view that the guidance of teachers can help learners clarify their learning goals, which will promote their direct involvement in relevant content-learning activities. In addition, due to the facilitator’s suggestions and support, the needs of learners can be met, thereby promoting their continuous social and cognitive participation [65]. Moreover, under the instructions of the observer, it is possible to increase the learner’s investment in perspective clarification and justification provision.

With respect to the detection of learners’ emergent roles (RQ2), the results show that many learners participate inactively in online co-reflection by playing a regular or peripheral role. A similar situation can be found in Ouyang and Chang’s [23] investigation where peripheral and regular students made few contributions to knowledge inquiry. However, combined with the findings of RQ1, we argue that the low-level participation of individuals does not necessarily determine the low-level co-reflection of the group. The reason for this can be, according to the results of the word cloud analysis (in Section 4.1), that the learners’ community identification may be an important factor affecting their co-reflection. This argument was also made by Clarà, Kelly, Mauri, and Danaher [2]; that is, a sense of belonging is a prerequisite for collaborative reflection in the online learning community. A stronger sense of community and social presence helps to alleviate learners’ feelings of isolation and help them to construct shared knowledge [3]. However, further exhaustive exploration is still needed.

In addition, referring to the interaction patterns among different co-reflection level learners (RQ3), more ambitious information about when and which roles learners tend to interact with can be obtained. Learners with low co-reflection levels pay more attention to interaction with teachers rather than peers, which may be the reason for their poor performance. As Clarà et al. [2] pointed out, some learners lack trust in their peers and are reticent about sharing their knowledge with others, and so they do not benefit much from doing so. Moreover, these learners may be afraid of the authority of teachers and scared of making mistakes or being criticized [3]; thus, their co-reflection is poor. Drawing on the achievement goal theory, we infer that learners with a low co-reflection level may have avoidance goals. They tend to conceal their worst performance through avoidance behavior [66]. Thus, if they doubt the adequacy or popularity of what they have to say, they may choose not to contribute. Furthermore, although they respond to the starter and mediator after interaction with teachers, it is difficult to find evidence that they engage in self-thinking; thus, we assume that their interaction with others may be attributed to face culture [67].

Unlike low-level learners, middle-level learners care more about the voice of their peers but are mainly limited to the influencers. Moreover, they do not rethink the interaction unless the observer asks them to give a detailed explanation. There are several explanations for this. First, according to the results of the word cloud analysis (in Section 4.1), middle-level learners may adopt a task-based goal to pay more attention to task completion rather than self-improvement [66]. Second, considering that they benefit more from influential participants but engage in less self-contribution, we argue that these learners possibly have a confidential attitude towards their knowledge [68]. Therefore, to a certain degree, the progress of their collaborative reflection is hindered. Third, for the differences in interaction sequences between influencer and other roles (e.g., starter, peripheral), a possible explanation is that middle-level learners may take one’s reputation as a criterion for judging whether others are trustworthy and cooperative [69].

Additionally, although high-level learners are focused on the opinions of influencers, which is different from middle-level learners, they engage in more positive self-thinking and place more emphasis on the voices of other peers (e.g., starters, peripherals, and mediators) before self-reflection. This illustrates that they might judge each other by their contributions rather than their reputation [69]. Moreover, given that they interact with multiple roles using collective discourse, we deduce that they may pursue group goals or self-improvement [66]. Consequently, they will open themselves up to accepting and sharing more ideas, thereby fostering their co-reflection.

## 6. Conclusions

Given the diverse roles and uneven co-reflection levels among learners in an online learning community, this research focused on exploring the interaction patterns of learners’ co-reflection levels and their emergent roles. A deep learning technique integrating ALBERT and LSTM was used to assess each learner’s co-reflection level, and SNA was employed to identify each learner’s emergent role. According to our findings, even if learners with different co-reflection levels had subtle differences in collaborative discourse, and although most of them took on an inactive role, thinking together can enable them to reach a higher level of collective reflection. Additionally, a series of LSA and chi-square analyses were conducted. The results revealed that higher-level learners had more proactive interactions with various roles and engaged in more self-thinking during the interaction.

This study makes several contributions and offers several practical implications. First, this study investigated the dynamics of roles in online co-reflection. Second, the current study is one of the few works that comprehensively considered the roles that the teacher and the learner play in co-reflection. Thus, this study contributes beneficial insights to facilitate the understanding of learners’ communication from the perspective of their roles in the digital learning environment. Third, the proven deep learning methods of ALBERT and LSTM methodologically contributed to larger-scale text classification tasks. In addition, one of the suggestions for online learning community practitioners and educators is that more opportunities should be given to learners to reflect collaboratively. Furthermore, some scaffolding should be provided to support learners’ deep participation from the perspective of their roles. Learners should be encouraged to assume collective responsibility and identity, value the voices of their peers, and actively share experiences and opinions. More importantly, it is recommended that teachers transform their roles and provide appropriate guidance, evaluation, and support to learners. More specifically, an influential learner should be organized to be the opener to motivate other learners to engage. In this process, the teacher is advised to act as an observer who asks learners to give detailed explanations or evidence. Subsequently, the teacher should take on the role of the guide to make an evaluation of or pose further requests to the learner. Other learners should be encouraged to perform the role of starters to share their new views and experiences based on the author’s contribution. It should be noted that although dynamic scaffolding is suggested, the responsibility of each role should be made clear in order to avoid ambiguity and conflict in communication.

Finally, it is necessary to point out the limitations of this study and to make suggestions for further research in this area. Although this study deepens the understanding of learners’ online co-reflection from the perspective of roles, a larger number of participants should be solicited in order to obtain more generalized results. The method of emergent role identification employed in this research is firmly based on previous works. The semantic discourse information of different roles is ignored, so an ambitious role positioning method can be considered in the future. Additionally, this research mainly focuses on revealing the interaction sequence of different roles but lacks a detailed analysis of the interactive strategy and discourse. Moreover, possible reasons why learners adopt different interaction patterns are explained based only on existing research and theory. Therefore, further research combined with qualitative analysis methods can be carried out to provide more detailed explanations about the interaction mechanism and motivation of learners with different co-reflection levels. For example, an interview method can be used to understand the underlying factors that affect the differences in learners’ interaction patterns. Finally, based on the findings of this study, designing the scaffolding for the learners’ co-reflection and examining its effects would be meaningful.

## Figures and Tables

**Figure 1 ijerph-19-02245-f001:**
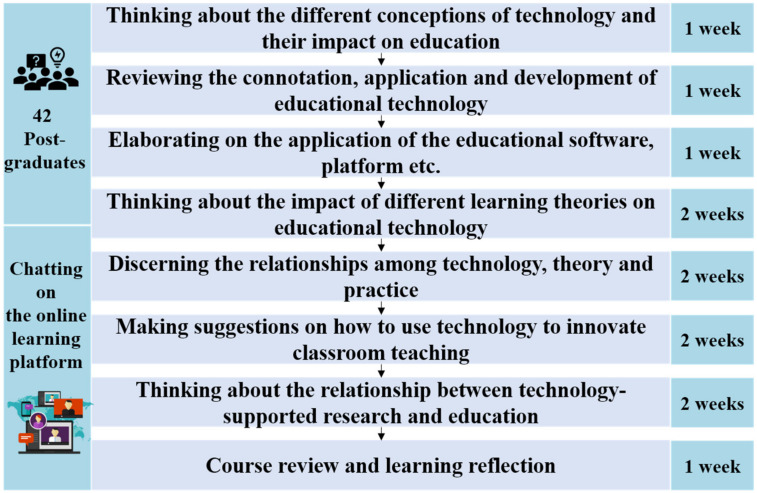
Co-reflection topics and process in an online learning community.

**Figure 2 ijerph-19-02245-f002:**
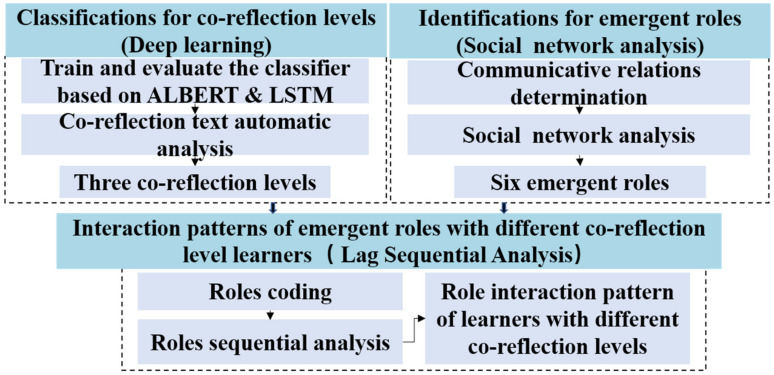
The data analysis process.

**Figure 3 ijerph-19-02245-f003:**
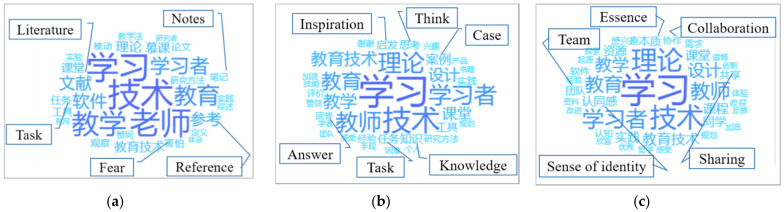
Word clouds for three co-reflection level learners in an online learning community. (**a**) is the word cloud of low-level learners; (**b**) is the word cloud of middle-level learners; and (**c**) is the word cloud of high-level learners.

**Figure 4 ijerph-19-02245-f004:**
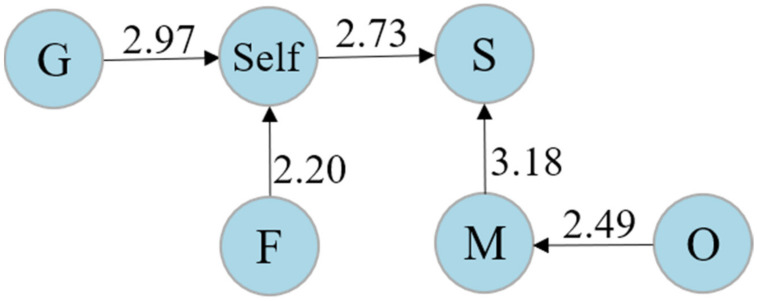
The transition diagram for learners at the low co-reflection level.

**Figure 5 ijerph-19-02245-f005:**
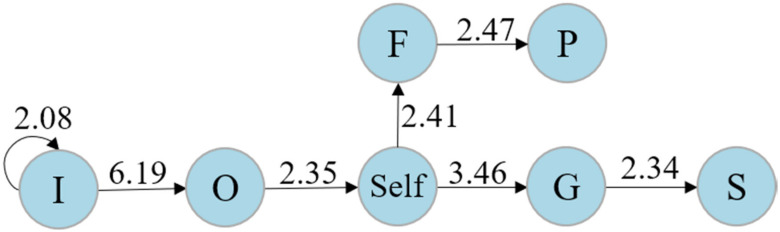
The transition diagram for learners at the middle co-reflection level.

**Figure 6 ijerph-19-02245-f006:**
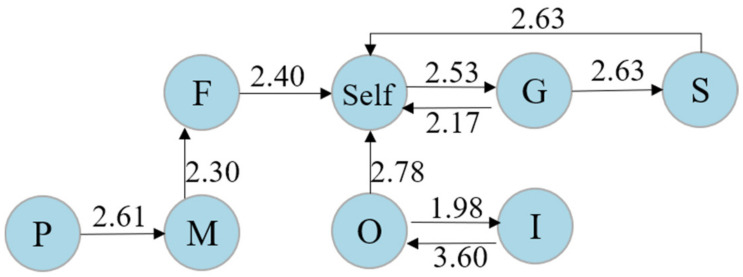
The transition diagram for learners at the high co-reflection level.

**Table 1 ijerph-19-02245-t001:** The coding scheme of the co-reflection levels.

Categories	Descriptions	Examples
1. Listing and Copying	Lists notes without explanations; copies information from or repeats other’s notes in a very close way.	I have learned from the materials recommended by the teacher and benefited a lot from the development of educational technology.
2. Brief Summary	Summarizes a few notes briefly and often incompletely.	Based on the views of most scholars, the procedure of design-based research is summarized as follows: 1. Analyze the problem. 2. Design the research proposal. …
3. Interpretation or Elaboration	Interprets the information on others’ notes with different wording or extend information using examples or evidence.	As you said, I have to admit that robots can do a lot of things that humans cannot…Additionally, I think they can also interact with students emotionally... The above are some supplements based on your opinion.…
4. Question-Based Discussion	Sees the discussion as question-based and a deepening process of seeking answers to questions.	In response to the issue you mentioned, I think the following questions should primarily be deliberated upon: 1.Why is this theory proposed? 2. What problem does it solve? …
5. Constructive Use of Information	Uses information, either from experts, books, the internet, or other related courses, life experience, etc., to justify or deepen ideas.	Once I heard a story in class... It can be seen that the teacher-centered model optimizes learning by imparting lots of knowledge to students, whereas the student-centered mode improves learning by providing students with opportunities for collaboration…
6. Intertwined Question Explanation	Keeps asking related questions, expresses doubts or seeks clarification; responses and explanations are intertwined progressively in the discussion.	Although MOOC has many advantages, as you mentioned, we all know that it still has many problems, the most prominent of which is...So, how should teachers design and organize learning activities? I think the following solutions can be considered: 1. The development of video… 2. Other related resources should be provided…
7. Meta-Cognition	Reflects on what the class does not know; realizes high points in the discussion; self-defines goals and tasks for exploration.	In general, we made a brief plan. First, preview the resources provided by the teacher… Second, search for relevant information from the Internet, and then... the purpose is to understand the development of learning theory and educational technology. The next plan is to....
8. Meta-Theory	Focuses on theories while developing the discourse; uses theories/conjectures to explain the phenomena, even with attempts to create new theories.	Why education has not reformed with the advancement of technology is worthy of our thinking…Although Diana Laurillard affirmed the value of technology in education in her works, the key is what problem technology solves... In general, we should slow down and focus on real problems... This may bring us a new picture of education.
9. Meta-Conversation	Focuses on examining what the discourse is about, especially reflecting on discourse goals; adopts a “we” perspective to assume collective responsibility for advancing knowledge; tackles difficult/important issues which may be neglected by the community.	Learning is a complicated process. We cannot only learn knowledge from books but also from others, because everyone has different experiences, methods, and viewpoints... What we have to learn is various ways of thinking...
10. Other	Some posts include greetings, thanks, simple compliments, etc.	Thanks! Come on! Excellent! etc.

**Table 2 ijerph-19-02245-t002:** The metrics of the identification of learners’ emergent roles [23].

	Participation		Influence		Mediation
	Outdegree	Outcloseness	Indegree	Incloseness	Betweenness
Leader	H or M	H or M	H or M	H or M	H or M
Starter	H	H	L or M	L or M	L or M
Influencer	L or M	L or M	H	H	L or M
Mediator	L or M	L or M	L or M	L or M	H
Regular	M	M	M	M	M
Peripheral	L or M	L or M	L or M	L or M	L or M ^1^

^1^ The range is set to L when a centrality score ranks between 0% and 20%, M when a score ranks between 21% and 80%, and H when a score ranks between 81% and 100%.

**Table 3 ijerph-19-02245-t003:** The effects of co-reflective text classification.

	Precision	Recall	Accuracy	F1
TF-IDF	58.67	58.79	59.73	58.73
Word2vec and RF	63.64	64.28	63.82	63.96
Word2vec and LSTM	67.85	68.42	68.69	68.13
BERT and RF	66.54	67.16	68.33	66.85
BERT and LSTM	75.01	74.76	75.03	74.88
ALBERT and LSTM	77.64	76.32	77.57	76.97

**Table 4 ijerph-19-02245-t004:** Descriptive results of the emergent roles in the online co-reflection process.

	Leader	Influencer	Mediator	Starter	Regular	Peripheral	Total
Frequency	6	3	3	2	13	12	39
Percentage	15.38	7.69	7.69	5.13	33.33	30.77	100

**Table 5 ijerph-19-02245-t005:** Comparison of SNA indicators for six emergent roles.

		Leader (L)	Influencer (I)	Mediator (M)	Starter (S)	Regular (R)	Peripheral (P)
Indegree	M (S.D)	45.33 (6.29)	56.33 (4.03)	22.67 (6.60)	13.00 (1.00)	18.92 (6.73)	7.08 (3.99)
	*p* value	0.000 ***					
	Post hoc test	L > M ***; L > S ***; L > R ***; L > P ***; I > L *; I > M ***; I > S ***; I > R ***; I > P ***; M > P ***; R > P ***					
Incloseness	M (S.D)	70.06 (3.52)	73.26 (1.88)	59.84 (4.92)	57.37 (1.20)	58.01 (3.78)	48.68 (5.77)
	*p* value	0.000 ***					
	Post hoc test	L > S **; L > M **; L > R ***; L > P ***; I > M **; I > S **; I > R ***; I > P ***; S > P *; M > P **; R > P ***					
Outdegree	M (S.D)	57.17 (5.73)	34.00 (5,72)	43.67 (3.30)	65.00 (1.00)	39.08 (9.43)	15.08 (8.76)
	*p* value	0.000 ***					
	Post hoc test	L > I **; L > M *; L > R ***; L > P ***; S > I ***; S > M *; S > R ***; S > P ***; I > P **; M > P ***; R > P ***					
Outcloseness	M (S.D)	66.34 (1.57)	58.63 (1.78)	63.73 (0.46)	72.62 (1.93)	61.10 (4.53)	51.46 (5.40)
	*p* value	0.000 ***					
	Post hoc test	L > I *; L > R *; L > P ***; S > I **; S > M *; S > R **; S > P ***; I > P *; M > P ***; R > P ***					
Betweenness	M (S.D)	57.50 (10.96)	25.56 (7.93)	54.49 (6.63)	36.73 (0.40)	16.17 (9.40)	1.01 (1.24)
	*p* value	0.000 ***					
	Post hoc test	L > P ***; M > P **^,1^					

^1^ * *p* < 0.5, ** *p* < 0.01, *** *p* < 0.001.

**Table 6 ijerph-19-02245-t006:** The sequential analysis results for learners with a low co-reflection level.

	Guide	Facilitator	Observer	Leader	Starter	Influencer	Mediator	Regular	Peripheral	Self
Guide	−0.38	−0.62	−0.38	−0.22	−0.68	−0.85	−0.62	−0.08	−0.27	2.97 *
Facilitator	−0.59	−0.96	−0.59	0.28	−1.06	0.68	0.35	−1.07	−0.42	2.20 *
Observe	−0.24	−0.38	−0.24	−0.81	−0.42	−0.53	2.49 *	1.00	−0.17	−0.59
Leader	0.76	0.70	0.76	0.73	−0.58	0.48	−0.35	−1.44	1.67	−0.69
Starter	−0.38	−0.62	−0.38	0.87	−0.68	0.57	1.24	−0.08	−0.27	−0.96
Influencer	−0.38	1.24	−0.38	−0.22	−0.68	−0.85	−0.62	1.05	−0.27	0.35
Mediator	−0.34	−0.55	−0.34	1.25	3.18 *	−0.76	−0.55	−1.07	−0.24	−0.85
Regular	0.94	−0.15	0.94	−0.70	−0.37	−0.03	−0.15	1.60	−0.55	−1.16
Peripheral	−0.17	−0.27	−0.17	−0.57	−0.30	−0.37	−0.27	1.93	−0.12	−0.42
Self	−0.38	1.24	−0.38	−1.30	2.73 *	0.57	−0.62	−0.08	−0.27	−0.961

* *p* < 0.05.

**Table 7 ijerph-19-02245-t007:** The sequential analysis results for learners with a middle co-reflection level.

	Guide	Facilitator	Observer	Leader	Starter	Influencer	Mediator	Regular	Peripheral	Self
Guide	−0.94	−0.76	0.67	−0.95	2.34 *	−0.16	0.10	−0.84	1.00	1.62
Facilitator	−1.49	−2.41	−1.21	0.09	−1.13	0.23	−0.69	0.56	2.47 *	0.96
Observer	−0.81	−1.30	−0.66	−0.27	−0.61	−1.01	−1.26	0.52	1.64	2.35 *
Leader	−0.37	1.26	−1.47	1.79	−0.44	−1.59	1.51	0.55	−2.49	−0.48
Starter	−0.30	−0.48	−0.24	−0.91	−0.23	1.17	−0.46	1.48	−0.55	−0.56
Influencer	−0.20	0.00	6.19 *	−2.38	0.38	2.08 *	0.13	−1.41	0.68	−0.50
Mediator	0.35	1.77	−0.72	−0.18	−0.67	−0.59	−0.55	0.42	−0.91	0.48
Regular	0.04	0.28	−0.80	1.36	0.25	0.70	−0.39	−0.55	−0.13	−1.39
Peripheral	1.47	−0.72	−0.75	−0.38	0.85	−0.71	0.16	1.04	−0.31	−0.35
Self	3.46 *	2.41 *	−0.62	−0.61	−0.58	0.57	0.69	−0.72	−1.41	−1.431

* *p* < 0.05.

**Table 8 ijerph-19-02245-t008:** The sequential analysis results for learners with a high co-reflection level.

	Guide	Facilitator	Observer	Leader	Starter	Influencer	Mediator	Regular	Peripheral	Self
Guide	−0.49	−1.05	−0.43	−2.26	2.63 *	1.46	−0.11	−0.82	0.07	2.17 *
Facilitator	−0.91	−1.95	−0.81	−0.56	−0.25	−0.70	−0.89	0.65	1.35	2.40 *
Observer	−0.43	−0.93	−0.39	−1.33	−0.58	1.98 *	−1.00	−0.45	−0.91	2.78 *
Leader	−0.22	0.86	−1.14	1.14	−0.14	−1.25	−0.03	−0.29	−1.10	0.86
Starter	−0.37	−0.80	−0.33	−0.16	−0.50	−0.25	−0.86	−0.79	0.62	2.63 *
Influencer	0.42	0.65	3.60 *	0.26	−0.22	−0.64	−1.46	0.74	0.75	−1.61
Mediator	−0.59	2.30 *	−0.53	−0.21	0.57	−0.04	−0.53	0.63	0.58	−1.97
Regular	0.15	0.09	−0.93	−0.83	−0.51	1.57	1.44	0.38	0.20	−1.80
Peripheral	−0.52	−0.11	−0.46	1.63	0.85	−0.98	2.61 *	−1.04	−0.06	−1.72
Self	2.53 *	0.09	1.18	1.30	−0.93	−0.02	0.63	−0.16	−1.45	−1.741

* *p* < 0.05.

**Table 9 ijerph-19-02245-t009:** Chi-square analysis results of the interaction with different roles.

	Low	Middle	High	Total	χ^2^ Tests	
					χ^2^ Value	*p* Value
Guide	29 (16.29)	89 (50.00)	60 (33.71) *	178	47.65	0.000 ^1^
Facilitator	28 (10.85)	135 (52.33)	95 (36.82)	258		
Observer	7 (8.75)	44 (55.00)	29 (36.25)	80		
Leader	37 (12.67)	144 (49.32)	111 (38.01)	292		
Influencer	7 (5.15)	64 (47.06)	65 (47.79) *	136		
Mediator	10 (14.49)	32 (46.38)	27 (39.13)	69		
Starter	4 (11.76)	8 (23.53)	22 (64.71) ^#^	34		
Regular	26 (10.00)	144 (55.38)	90 (34.62)	260		
Peripheral	2 (2.27)	43 (48.86)	43 (48.86) *	88		
Self	4 (4.35)	40 (43.48)	48 (52.17) *	92		

* represents a statistically significant difference for low co-reflection level learners; ^#^ represents a statistically significant difference for middle co-reflection level learners.

## Data Availability

The data are not publicly available due to privacy restrictions.

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
