# Peer review of "Identifying Learners’ Interaction Patterns in an Online Learning Community"

_ijerph, 2022, doi:10.3390/ijerph19042245_

Round 1
Reviewer 1 Report
It would be helpful to define what is meant by "levels" in relation to collaboration in the introduction-- is it quantity of reflection? Quality? By what measure? Might also help to identify here whether the reflections are discussion posts, papers, journal entries, etc. This is addressed in the methods section, but a brief explanation in the introduction would be helpful.
In the literature review, page 2 lines 75-76, the author writes "this study maintains..." Are they referring to the study being reviewed? Or their own? If it is their own, and they are making a hypothesis, that should be made more clear, and probably moved to the methods section.
The literature review refers often to the various roles that people might be assigned or adopt in discussions, but does not discuss what these roles are or give examples. In the methods section, they introduce the terms guide, facilitator and observer saying that the importance of these roles for teachers has already been established. However, the literature review talked about teachers roles generally, and did not address these roles specifically. The author should identify the roles that students might develop or be assigned with some examples, and establish the teachers' roles more clearly in the literature review.
In section 3.2.1, page 6, line 255, the authors suddenly indicate that an additional set of data of online posts was added to the original data set. While the participants and conditions for the original data set were explained in some detail, the second set is not. Adding in data could change the nature of the study and impact results, so a more detailed explanation of this data set is needed. Also, it is unclear why this data set was added in at all. The author states that the original set was "insufficient"-- but how so? Why and how was it determined that more data was needed?
It would be helpful to define the metrics of outdegree, outcloseness, indegree, incloseness and betweenness-- what characteristics are these measuring, and what do they tell us?
Reviewer 2 Report
Title
The title is too long.
My suggestion for the title is “Identifying Learners' Interaction Patterns in An Online Learning Community.”
Introduction
There are several different backgrounds explained in the literature review section. It is better to explain this part, why this research must do, in the introduction section. Furthermore, in the literature review section it is described the position of this research in completing or answering problems that are still gaps in several existing literature and has been reviewed and analyzed by the author. For more details, you can follow my comments, such as:
Lines 30-47,
There is three “However” words in this paragraph. I think you can divide this paragraph to be 3 paragraph to make sure that every paragraph has one main idea that you want to explain.
Lines 42 – 43,
In the previous paragraph, authors discuss about collaborative behavior and reflection. In this sentence, my suggestion is necessary to explain the relationship between student participation and perception with the previous discussion.
Lines 45 – 46,
This sentence is confusing in the paragraph. Please correct it.
Lines 60 – 63,
In this section, it is better to write about the implications and significant contributions of this research so that readers can know the importance of reading or learning about the content written by the author. In addition, emergent roles and collaborative reflection can be explained and discussed in this section if the title is omitted to make the title more effective.
Literature Review
In the literature review section, important points should be explained in the introduction section.
Research Method
Several important points need to be supported by arguments, such as in lines 228-229, why the Liru Cloud Classroom platform was chosen to learn from the many existing platforms. In addition, lines 253-254 explain when reflective text data is also collected from other courses whether it will not be biased.
Discussion
The discussion section has been explained the results of the research and why this could happen. However, the explanation of the factors that influence the findings is still in the form of 'probability' from the results of previous studies. It would be more in-depth to the research subject, for example, from direct interviews with research subjects to find out the fundamental factors that influence some points of research findings, so that the discussion results can be more comprehensive. Or if it is not possible, then this deficiency can be written in the study's limitations.
Round 2
Reviewer 2 Report
My comments have fulfilled the content of the revised paper. However, the level of English throughout your manuscript does not meet the journal's required standard. To help with English language usage and quality, we strongly recommend that you consult with a colleague whose native language is English or use a professional language editing service.
